# Short-Term Consequences of Single Social Defeat on Accumbal Dopamine and Behaviors in Rats

**DOI:** 10.3390/biom13010035

**Published:** 2022-12-24

**Authors:** Vsevolod V. Nemets, Alex L. Deal, Vladislav E. Sobolev, Vladimir P. Grinevich, Raul R. Gainetdinov, Evgeny A. Budygin

**Affiliations:** 1Institute of Translational Biomedicine and Saint Petersburg University Hospital, Saint Petersburg State University, Saint Petersburg 199034, Russia; 2Department of Neurobiology and Anatomy, Wake Forest School of Medicine, Winston-Salem, NC 27157, USA; 3Laboratory of Comparative Biochemistry of Enzymes, Sechenov Institute of Evolutionary Physiology and Biochemistry of the Russian Academy of Sciences, Saint Petersburg 194223, Russia; 4Department of Neurobiology, Sirius University of Science and Technology, Sochi 35340, Russia

**Keywords:** social defeat, behavior, dopamine release, dopamine D2 autoreceptor, dopamine depletion, voltammetry

## Abstract

The present study aimed to explore the consequences of a single exposure to a social defeat on dopamine release in the rat nucleus accumbens measured with a fast-scan cyclic voltammetry. We found that 24 h after a social defeat, accumbal dopamine responses, evoked by a high frequency electrical stimulation of the ventral tegmental area, were more profound in socially defeated rats in comparison with non-defeated control animals. The enhanced dopamine release was associated with the prolonged immobility time in the forced swim test. The use of the dopamine depletion protocol revealed no alteration in the reduction and recovery of the amplitude of dopamine release following social defeat stress. However, administration of dopamine D2 receptor antagonist, raclopride (2 mg/kg, i.p.), resulted in significant increase of the electrically evoked dopamine release in both groups of animals, nevertheless exhibiting less manifested effect in the defeated rats comparing to control animals. Taken together, our data demonstrated profound alterations in the dopamine transmission in the association with depressive-like behavior following a single exposure to stressful environment. These voltammetric findings pointed to a promising path for the identification of neurobiological mechanisms underlying stress-promoted behavioral abnormalities.

## 1. Introduction

Stressful events provoke a chain of biochemical and neurochemical responses in the body and brain [1,2,3,4,5,6,7]. Some of these changes can be protective or adaptive in nature, while other may play a significant role in the development of dysfunctions followed by the progression to diseases [1,3,5,8,9,10,11,12]. The stressed-derived aberrations in the dopamine transmission have received particular attention due to the involvement of this neurotransmitter in pathological conditions, such as drug and alcohol addictions [8,13,14], anxiety, depression [15,16], and post-traumatic stress disorder (PTSD) [17,18,19,20]. In fact, these psychopathologies may be significantly escalated by chronic as well as by single exposure to stress [3,21,22]. Insights into the neurochemical responses to stress can be gained from valid animal models. Such a translational model, which has the etiological relevance for examining stress-related pathological conditions, is a social defeat [3,12,23,24,25,26,27,28]. The applicability of this paradigm is exceptionally simple given the almost ubiquitous presence of territorial or hierarchical aggression in males [6]. The interaction of the tested male with an aggressive subject results in increased heart rate, blood pressure, an elevation of body temperature and a powerful endocrine response [7,29,30]. These physiological changes in combination with evoked defensive and submissive behaviors that are clearly observed in a defeated animal indicate an acute stress condition. During exposure of an intruder rat to the social defeat procedure, the burst firing of the VTA dopamine neurons is increased [31], and as a consequence, dopamine release is enhanced in terminal fields including the nucleus accumbens [31,32].

To reproduce the phenotype of stress-related psychopathologies, repeated social defeat was predominantly employing [3,6,26,27,28,33,34]. Such a paradigm is more equivalent to the current theories on the development of depression in human rather than a single acute stress [35]. On the other hand, a strong acute trauma might trigger conditions such as a PTSD. The mechanisms of the development of this pathology are still unclear and social defeat model can be useful in this regard. In fact, robust and relatively long lasting behavioral and physiological changes can be elicited even by a single exposure to social defeat [3,6,36]. According to the study performed by Meerlo and colleagues, a profound increase in body temperature during the circadian resting phase along with changes in spontaneous home cage activity of rats can continue for up to 10 days after social defeat [29]. The same work revealed that a food intake and growth in such animals were suppressed for a number of days, whereas the decrease in locomotion under a novel environment was evident 2 days after stress [29]. Currently, it is well known that the exposure to a single acute stressor induces neuroadaptations in VTA dopamine cells [3]. Thus, the induction of long-term potentiation (LTP) at GABA_A_ synapses on VTA dopamine neurons was blocked 24 h following acute exposure to stress [37]. Since dopaminergic transmission is under a potent control of VTA GABA neurons [38,39], stress-triggered loss of LTP is expected to increase dopamine release in the terminal field [37]. In agreement with this speculation, electrophysiological recordings revealed that burst firing of VTA dopamine neurons increases with acute restrain stress and persists at least for 24 h [40]. It was also speculated that the loss of LTP at inhibitory synapses in combination with the induction of LTP at excitatory synapses can result in enhanced responsivity of VTA dopamine neurons to future stimulations [3]. However, direct evidence that increased dopamine release is a consequence of single stress is currently lacking.

Therefore, the present study was designed to explore the consequences of a single social defeat on accumbal dopamine release evoked by an electrical stimulation of the VTA measured by a fast-scan cyclic voltammetry (FSCV). Based on the results of previous studies [37,40], neurochemical measurements as well as behavioral assessments performed on separate group of rats, were made 24 h following the stress procedure.

## 2. Materials and Methods

### 2.1. Animals and Behavioral Tests

Male Sprague Dawley (300–350 g) and Long Evans (450–500 g) rats were housed on a 12 h light/dark cycle with food and water available ad libitum. All animal procedures were conducted in accordance with the National Institutes of Health Guide for the Care and Use of Laboratory Animals. All experiments conformed within international guidelines on the ethical use of animals. Figure 1A shows a schematic representation of the experimental design.

For social defeat procedures, a Sprague Dawley rat was placed into the home cage of the resident (Long Evans) in order to create a robust stressful environment for the intruder [6,41,42]. During the first and last 5 min segments of the experimental session (20 min), the resident and intruder rats interacted through a wire mesh inset cage, which permitted only a visual, auditory, and olfactory perception of an aggressive subject (Figure 1B). The second segment was designated for direct physical contact between resident and intruder. During this 10 min period, the protective inset cage was removed. In a separate set of experiments (control), a Sprague Dawley male was reintroduced to its familiar cage mate for 20-min. Since these rats (the same strain, age and size), were housed together for several weeks, aggressive confrontations during their interactions were prevented. In fact, behavioral elements such as biting attacks, aggressive postures and pursuit were not observed in these animals. Therefore, they could serve as a control group.

Each 20 min social defeat session and control group interactions were video recorded. Behavioral patterns were recognized as follows: defensive, freezing (rat does not move), explorative (rearing, sniffing), running, submissive (rat lying on its back) and other (walking, standing). Duration of each behavioral element was assessed in both socially defeated (SD) and control rats. Every effort was made to minimize suffering and reduce the number of animals in this study. Rats (including control) were tested in the standard battery of behavioral tests 24 h following social defeat and social interaction paradigms.

Sucrose preference test was performed to reveal whether an anhedonic condition could be triggered by short-term stress exposure. A two-bottle choice design was used where the rats had access to water and a 10% sucrose solution 18 h/day for 3 days/week over 2 weeks. At the beginning and end of the drinking period the bottles were weighed and a fluid intake was calculated. No previous food or water deprivation was applied before the test. The intake of water and 10% sucrose was measured during the dark circadian phase [43]. Final measurements were taken 24 h after the social defeat or social interaction (control condition) sessions. Sucrose preference was calculated as a ratio of the consumed 10% sucrose solution (g) to the sum of total fluid intake (sucrose + water, g) × 100% [44].

An open field test was used to explore short-term consequences of a single bout of social defeat on locomotion and stereotypical behaviors. Rats were acclimated to the testing room and then individually placed at the center of the plastic arena (40 × 40 × 40 cm). Their spontaneous activity was registered) for 6 min with a video-recording system (EthoVision XT 11.5, Noldus. Wageinen, The Netherlands). Between subjects, the arena was cleaned with 3% hydrogen in order to eliminate olfactory cues. The duration of behavioral patterns (freezing, rearing, grooming, etc.) was counted manually by analyzing the recorded videos [45].

Finally, the forced swim test procedure [35,46] was performed to verify a stress-like phenotype in rats 24 h after social defeat. Briefly, animals were allowed to acclimate to the testing room for 40 min. Then, each subject was placed in a water-filled cylindrical glass container (height–45 cm, diameter–28 cm) with the water temperature 23 ± 1 °C. The experimenter evaluated behavior for 6 min of the test as either swimming or floating (immobile). The floating behavior was determined as the nonappearance of any directed movements of the body or head [4]. The container was washed with soap and hot water after each subject.

### 2.2. FSCV Recordings of Dopamine Release in Rat Nucleus Accumbens

Electrically evoked dopamine release was measured in the nucleus accumbens (NAc) of anesthetized rats 24 h after the social defeat procedure. Rats were anesthetized by using a single intraperitoneal (i.p.) injection of urethane (1.5 g/kg) and secured in a stereotaxic frame. Holes were drilled to allow for the implantation of electrodes into the brain. A stimulating electrode was inserted into the VTA (AP: −5.2 mm, ML: 1.0 mm DV: −8.4 mm), a carbon fiber working microelectrode (exposed fiber length 75–100 µm; diameter 6 µm) was placed into the NAc (AP: 1.3 mm, ML: 1.3 mm, DV: −7.1 mm) and an Ag/AgCl reference electrode into the brain tissue of the contralateral hemisphere. The electrodes were connected to the voltametric amplifier interfaced with a computer running the specialized software. To explore the difference in frequency-dependence of dopamine release between SD and control rats, 1-s electrical stimulations (330 µA) of the VTA were made at different frequencies (5, 10, 20, 30, 50 and 60 Hz) every 10 min. For the pharmacological study, following acquisition of a stable baseline dopamine signal (3 signals with amplitudes within 10% variability), raclopride (dopamine D2 receptor antagonist) at 2 mg/kg (i.p.) dose was injected. This raclopride dose robustly affects dopamine transmission at the level of presynaptic terminals in a way that allows for reliable detection of increased electrically evoked dopamine release with FSCV [5,47]. Electrically evoked (60 Hz, 60 pulses) dopamine effluxes following raclopride injection were measured every 10 min for 90 min.

The depletion protocol included three consequent stimulations (330 µA, 60 Hz, 600 pulses), which were made with 1–2 s intervals. Then, a regular stimulation (330 µA, 60 Hz, 60 pulses) was employed with 1 min (14 stimulations), 5 min (3) and 10 min (5) intervals to allow dopamine signal recovery.

Extracellular dopamine was detected at the carbon fiber electrode every 100 ms by applying a triangular waveform (−0.4 V to +1.3 V and back to −0.4 V vs. Ag/AgCl, 400 V/s). The dopamine signal was verified by a background-subtracted cyclic voltammogram characterized by oxidation and reduction peaks occurring at +0.6 and −0.2 V, respectively [47,48,49].

### 2.3. Histology

To verify the placement of the detecting electrode animals were overdosed with chloral hydrate (1000 mg/kg) and perfused intracardially with 0.9% saline. The brains were rapidly removed, and brain samples were fixed in a buffered 10% formalin solution for 24 h, then were washed in water for 20 min and kept at +4 °C in 30% sucrose in 0.1 M phosphate buffer for 48–96 h as previously described [50]. Next, brain samples were frozen in a cryostat chamber after cryoprotectant treatment with a 7% sucrose solution in 0.1 M phosphate buffer. Brain sections at a 15 micron thickness were obtained using a Slee MEV cryostat (SLEE Medical, Germany) at −18 °C. The sections were stained with hematoxylin-eosin according to a standard protocol. Microscopy of the sections was performed using an Axiostar plus microscope (Carl Zeiss, Germany).

### 2.4. Statistical Analysis

Data for each behavioral element was accounted as a percent of total observation time. Voltammetry measurements (current, nA) were converted into a concentration of released dopamine and expressed in molar (M) or as % of preceding background values. Statistical analysis of voltammetric data was performed by applying a repeated-measures two-way ANOVA. For behavioral data D’Agostino and Pearson omnibus normality test was used for counting whether the values followed Gaussian distribution and then in case of normal distribution we used unpaired two-tailed *t*-tests, otherwise nonparametric Mann–Whitney U test was performed. All analysis was carrying out using GraphPad Prism (version 6.05, San Diego, CA, USA). The data are expressed as a mean ± SEM with a criterion for significance set at *p* ≤ 0.05.

## 3. Results

### 3.1. Behavioral Changes Observed during a Single SD Session

The present study revealed evident behavioral alterations in the “intruder” rats during SD exposure. In the paradigm (Figure 1 and Figure 2), the average time for a direct physical contact during the rival SD experience was 436 ± 81.5 s. This time varied depending on the degree of violence (attacks with bites) of the resident rat. On average, the defeated intruder rats were subjected for 2.8 ± 0.4 s to aggressive clinch attacks concurrently with bites for 0.4 ± 0.2 s. A latency period for the first attack was detected within 1.3 ± 0.2 s. The defeated rats exhibited (Figure 2A) active behaviors such as a defense (12.1 ± 3.0%), running (0.7 ± 0.2%) and exploration (26.6 ± 5.8%), as well as conspicuous passive behaviors including freezing (25.6 ± 4.6%) and submissive (1.7 ± 0.6%) behaviors. In control intruder rats (Figure 2B) while they were interacting with their counterparts, a freezing was barely noticed (0.1 ± 0.1%), whereas an exploration was not different (27.0 ± 4.1%). Moreover, grooming behavior was not observed in socially defeated intruder rats, while it was obvious in control animals. Figure 2C demonstrates a significant increase in freezing behavior along (Mann–Whitney U test, *p* = 0.0006) with a decrease in other behaviors, such as walking and standing (unpaired two-tailed *t*-test, *p* = 0.0001), with no significant difference in explorative behavior such a rearing and sniffing (27.0 ± 3.6 vs. 26.6 ± 5.1; *p* = 0.0955, unpaired two-tailed *t*-test) in defeated rats when compared to control rats.

### 3.2. Short-Term Behavioral Consequences Observed 24 h Following a Single SD Session

Several tests did not find significant alterations in behaviors of intruder rats following the SD experience. There were no differences between defeated rats and control in sucrose preference (84.2 ± 3.9% vs. 80.4 ± 4.9%, *p* = 0.8048, Mann–Whitney U test) in two-bottle choice test (Figure 3A). Similarly, behavioral measures in the open field test were not significantly different between groups (Figure 3B–E), including distance moved (27.6 ± 2.0 m vs. 26.1 ± 1.6 m, *p* = 0.5819, unpaired two-tailed *t*-test), time spent in the center of arena (53.7 ± 10.6 s vs. 43.4 ± 5.8 s, *p* = 0.2176, Mann–Whitney U test), rearing (63.9 ± 6.1 s vs. 50.7 ± 3.6, *p* = 0.1051, Mann–Whitney U test), grooming (9.8 ± 3.2 s vs. 14.2 ± 4.6 s, *p* = 0.2176, Mann–Whitney U test) and locomotor activity (123.1 ± 9.3 m vs. 129.0 ± 13.7 s, *p* = 0.8559, unpaired two-tailed *t*-test). However, socially defeated rats showed a significant increase in the immobility time during the forced swim test (0.8 ± 0.2 vs. 0.4 ± 0.1; *p* = 0.0427, unpaired two-tailed *t*-test, Figure 3F) when compared to controls.

### 3.3. Consequences of Single SD Session for Dopamine Dynamics in Rat Nucleus Accumbens

FSCV recordings coupled with electrical stimulation were used to evaluate dopamine release changes in defeated intruder rats 24 h following SD procedure. In Figure 4 the baselines of dopamine effluxes from the nucleus accumbens of control and defeated animals are shown following electrical stimulation of VTA (60 Hz, 60 pulses).

The amplitude of dopamine release was increased in both groups in frequency dependent manner (repeated measures two-way ANOVA; F (6, 78) = 15.59, *p* < 0.0001) (Figure 5). However, there was a significantly more pronounced dopamine response in defeated rats compared to control animals (repeated measures two-way ANOVA; F (1, 13) = 7.005, *p* = 0.0201). Sidak’s multiple comparisons test revealed a higher dopamine release in defeated rats vs. controls at higher frequencies (60, 50, and 40 Hz; *p* < 0.05) but not at lower frequencies (30, 20, 10 and 5 Hz; *p* > 0.05) (Figure 5).

Administration of the selective D2 dopamine receptor antagonist raclopride (2 mg/kg, i.p.) resulted in significant increase of electrically evoked dopamine release in both groups (repeated measures two-way ANOVA; F (8, 88) = 30.26; *p* < 0.0001), while a greater effect was observed in controls compared to defeated rats (F (1, 11) = 5.321; *p* < 0.05) (Figure 6).

Following the dopamine depletion protocol, both groups showed a similar dramatic reduction in electrically evoked dopamine efflux (repeated measures two-way ANOVA; F (21, 210) = 14.76; *p* < 0.0001) with no difference between groups (F (1, 10) = 1.141; *p* > 0.05) (Figure 7).

## 4. Discussion

The present study was mainly focused on the short-term consequences of a single social defeat on accumbal dopamine dynamics, while behavioral assessments were also achieved. Expectedly, robust stress-associated behaviors were evident in intruder rats during their confrontation with an aggressive resident. However, the majority of performed tests did not reveal significant alterations in behaviors of defeated rats 24 h following the interaction. Thus, a sucrose intake, locomotor activity, rearing and grooming were identical to those in control animals. Nevertheless, stressed animals demonstrated a substantial increase in the immobility time during the forced swim test. This social defeat consequence, which indicates a deficiency in natural escape behavior, was paralleled with changes in dopamine release evoked by an electrical stimulation of the VTA. Remarkably, the increased dopamine efflux was observed following the VTA activation at high frequencies (40, 50 and 60 Hz), while lowest frequencies (5, 10, 20 and 30 Hz) did not result in the differences in accumbal dopamine response between defeated and control rats. Furthermore, the administration of dopamine D2 receptor antagonist, raclopride (2 mg/kg, i.p.) caused a significantly weaker effect on electrically evoked dopamine release in stressed subjects compared to control. The use of the dopamine depletion protocol revealed no alteration of the reduction or recovery of the amplitude of dopamine release following stress triggered by a social defeat.

During social defeat in rodents, acute physiological adaptations, which include an activation of cardiovascular and endocrine systems [7,52] along with neurochemical responses, such as an increased dopamine and norepinephrine release [6,31,53], as well as striking changes in patterns of behavioral activity [6,31] are well recognized. Most of these changes are reduced or eliminated within 24 h after stress exposure. However, some behavioral and neurobiological effects can be determined even after a single episode of social defeat [6]. For example, single social defeat resulted in a significant decrease in locomotion of Tryon Maze Dull S3 rats in the open field test, which lasted for several days after a single exposure [54]. In contrast to these findings, no changes in locomotor activity, as well as in many other behaviors (e.g., sucrose intake, grooming, rearing and others) were observed in the present study. The strain difference (Sprague Dawley versus Tryon Maze Dull S3) and, more likely, the divergence in the social defeat procedure could be accounted for the discrepancy in results. Specifically, in the previous study rats were placed in the cage of an aggressive male for 1 h, while a total time of stress exposure in our experimental approach was 20 min. The longer confrontation between a resident and intruder also resulted in a reduction in the amplitude of the daily temperature rhythm, the profound increase in body temperature during the circadian resting phase, the decrease in spontaneous home cage activity and the decrease in locomotion in a novel environment [29,55]. Perhaps, a lengthier single defeat could trigger more behavioral and physiological consequences.

Nevertheless, the immobility time in the Porsolt forced swim test (FST), which is considered an indicator of a depression-like state [35,46], was affected in the current study. Though previous studies on a single social defeat did not focus on floating behavior [29,54,55], it is well documented that repeated social defeat prolongs an immobilization time in FST, indicating the development of helplessness and despair in animals. Remarkably, recent study in mice revealed a possible link between striatal dopamine turnover and floating behavior [4].

In the current study this behavioral consequence of social defeat was paralleled with changes in the accumbal dopamine release evoked by an electrical stimulation of the VTA. Noticeably, dopamine release was enhanced under the condition, when the stimulations were performed at high frequencies, which trigger phasic pattern of dopamine neurotransmission [56,57]. The increased phasic dopamine signaling in rat nucleus accumbens was previously revealed during a social confrontation and following a defeated subject was returned into its home cage [31]. In agreement with these neurochemical changes, substantial increases in burst frequency were detected in the VTA dopamine firing patterns during an aggressive confrontation as well as in home cage [31]. Furthermore, it was found that neurons with higher burst levels under normal conditions (before social defeat procedure) did not switch from non-bursting to bursting types, whereas cells with higher burst activity displayed augmented increases in bursting [31]. This is consistent with the results on electrophysiological consequences of single restraint, which is also a highly stressful condition for conscious animals [40]. Moreover, the evidence was obtained that the shift to the increased burst firing persists 24 h after initial exposure to restraint stress. Because dopamine bursts result in enhanced efflux at synaptic terminals [31,58], increased phasic dopamine signaling should be predictable under this circumstance. However, no neurochemical exploration of changes in fast dopamine transmission was performed in this earlier study [40]. Together, these findings suggest that the increased phasic dopamine response that develops during a single stress exposure can be preserved at least for 24 h after the termination of stress. Perhaps, this consequence may disrupt normal dopamine dynamics in the nucleus accumbens and therefore endorsing some depressive-like behaviors.

A single stress promotes several neuroadaptations, which can be responsible for observed dopamine changes. Thus, acute stress-induced blockade of the LTP induction at GABA_A_ synapses on VTA dopamine neurons [37,59] may result in enhanced responsiveness of dopamine cell bodies. The same outcome can be reached due to the induction of LTP at excitatory synapses in the VTA. In fact, the increases in AMPA/NMDA ratio were triggered within 2 h of acute stress and have been continued at least 24 h [9,60]. These fast neuroplasticity alterations in the VTA, which independently predict an enhanced dopamine release in the nucleus accumbens, can be combined in response to an acute stressor [3].

Remarkably, the consequences of social defeat on dopamine dynamics can be evident in an ex vivo preparation, where the release is evoked by electrical stimulation of accumbal slices, which have no connectivity from the dopamine cell body region (VTA) [61]. In fact, the magnitude of dopamine efflux was significantly greater 1 h after the last session of the 3-day stress exposure. These data suggest the development of neuroadaptations at the level of dopamine terminals. For example, the kappa-opioid receptors and their endogenous ligand dynorphin can be involved in adaptations to stress. There is evidence for the decrease in dynorphin mRNA levels in mouse nucleus accumbens following chronic social defeat, while acute stress leads to opposite consequence [62]. It was revealed that the activation of kappa opioid receptors, which are also located on dopamine terminals, suppresses accumbal dopamine efflux [63,64]. Consequently, the reduced inhibition of dopamine release via activation of these receptors could lead to the enhanced dopamine in the nucleus accumbens due to diminished level of endogenous agonist and contrary wise. This is in agreement with the results obtained in ex vivo preparation following repeated social defeat [61]. However, the data on dopamine release changes following the current paradigm do not fit with the expectation that was based on dynorphin changes after acute stress [62].

On other hand, corticotropin-releasing factor (CRF) can be responsible for the observed dopamine alterations. This neuropeptide releases in response to acute stress promoting adaptive and maladaptive behaviors [65,66] through its action on central CRF receptors [67]. Previous studies on brain slices revealed that, acting acutely within the nucleus accumbens core, CRF increases dopamine release [10,68,69]. Therefore, an emerging hypothesis can be that multiple circuit mechanisms, which are located at the level of dopamine cell bodies and terminals, may result in increased dopamine release in the nucleus accumbens. Though, the consequences of acute single trauma and chronic stress on accumbal dopamine transmission can be opposite.

Altered dopamine efflux and consequent changes in its extracellular concentrations potentially may lead to presynaptic neuroadaptations, which are directed to compensate nonphysiologocal changes. This may impact the processes of synthesis, autoreceptor regulation of neurotransmitter release and reuptake. For example, the acceleration in dopamine uptake presumably due to the persistently elevated dopamine concentration in dopamine terminals was found using different models of drug addiction [13,15].

To find whether the consequence of single social defeat on dopamine release is capable of affecting dopamine dynamics at the presynaptic level, we applied the dopamine depletion protocol that allows to reveal changes in synthesis, reuptake and autoreceptor regulation of dopamine [70,71,72]. This approach is based on the finding that a certain amount of time is required for the recovery of the evoked dopamine release from the effects of long electrical stimulations at high frequency. The dynamics of the decline and recovery of detected dopamine should reflect the overall regulation of presynaptic transmission. Despite the significant difference in dopamine release between control and stressed rats, the long stimulation protocol resulted in a full recovery of the depleted dopamine signal in the same levels of the depletion and following recovery of dopamine signal. Therefore, these data may suggest that the balance of intrasynaptic events in rat nucleus accumbens is not disturbed following single social defeat.

Nevertheless, single social defeat resulted in a weaker effect of raclopride on electrically evoked efflux compared to the control that might suggest alteration in D2 receptor regulation of dopamine release. Interestingly, there is an evidence that acute exposure to aggression increases D2 receptor density in the nucleus accumbens of the opponent rat [73]. However, these changes were observed immediately following the exposure session. Therefore, it should still be figured out whether or not they were preserved within 24 h. Moreover, the increased D2 receptor density in the nucleus accumbens would predict rather an enhanced dopamine release following D2 receptor antagonist than its decrease. Perhaps a more simple explanation of the altered effect of raclopride can be offered. Since raclopride and dopamine compete for the D2 receptors, increased electrically evoked dopamine could modify the pharmacological effect of the drug, weakening presynaptic autoreceptor blockade in the VTA-nucleus accumbens terminals and therefore down-regulating the release.

## 5. Conclusions

Our data demonstrated marked changes in the accumbal dopamine neurotransmission together with the indication of depression-like condition determined 24 h following the exposure to a social defeat. Remarkably, these neurochemical and behavioral changes occurred as consequences of only a single exposure to stressful event. Collectively, these findings pointed in a promising direction for the identification of the neuronal circuitry and neurochemical mechanism underlying stress-triggered development of psychopathologies such as PTSD and depression.

## Figures and Tables

**Figure 1 biomolecules-13-00035-f001:**
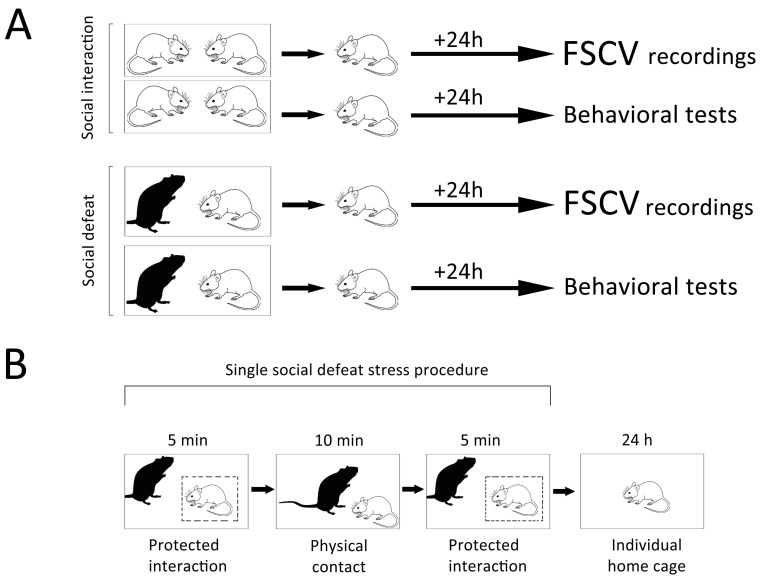
Schematic illustration of the experimental design. (**A**) Two groups of male Sprague Dawley rats (intruders) were exposed to different behavioral paradigms. In the first one (control condition), a subject was reintroduced to its cage mate (Sprague Dawley male), whereas in the second paradigm (social defeat), a rat was placed into the cage of an aggressive Long Evans male for the same period of time (20 min). 24 h after these procedures, intruders were subjected to either FSCV study or behavioral assessments (sucrose consumption, open field test and forced swim test). (**B**) During the first and last segments of the social defeat session, rats could interact through a wire mesh inset cage only. In the second segment of the paradigm, animals were allowed to physically contact with each other. Following this procedure, subjects were returned to their home cages for 24 h.

**Figure 2 biomolecules-13-00035-f002:**
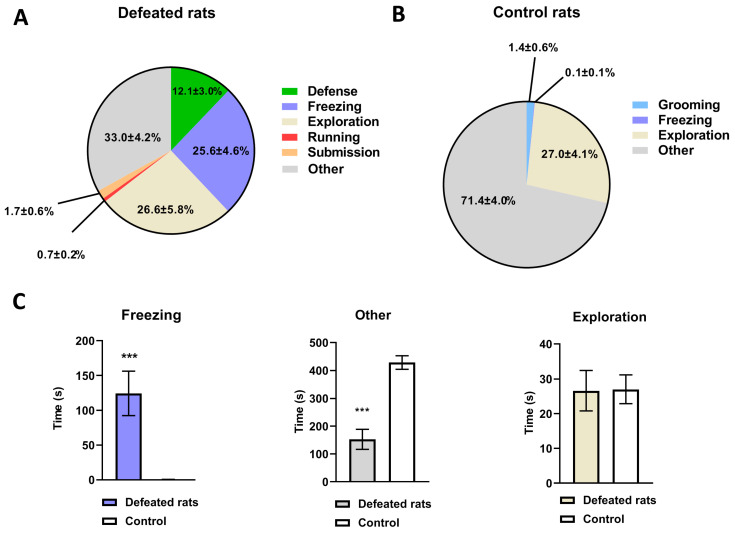
Rat behavior displayed during social defeat stress. Behaviors were characterized as defensive, freezing (rat does not move), explorative (rearing and sniffing), running, submissive (rat lying on the back), or other (walking, standing). (**A**)–Distribution of behaviors exhibited by intruder rats during social defeat stress presented as a percent of total session time exposure (n = 8), (mean ± SEM). (**B**)–Distribution of behaviors exhibited by intruder “control” rats during a 20 min exposure to a non-aggressive counterpart presented as a percent of total session time (n = 8), (mean ± SEM). (**C**)–Effect of social defeat stress on the display of select behaviors compared to non-stressed controls. Data presented as mean ± SEM, Unpaired two-tailed *t*-tests or Mann–Whitney U test was used, (n = 8 per group), *** *p* ≤ 0.0006.

**Figure 3 biomolecules-13-00035-f003:**
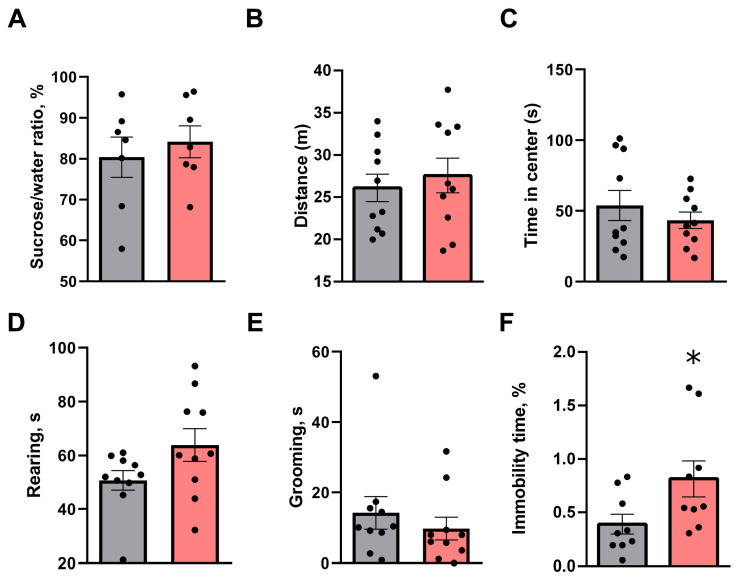
Behavioral consequences of single social defeat stress exposure. Rats were tested in the standard battery of behavioral assessments, including a two bottle choice, open field and forced swim tests. (**A**) Sucrose preference (n = 7 per group), (**B**–**E**) locomotor and anxiety-like behaviors (n = 10 per group) and (**F**) floating behavior (n = 9 per group) were evaluated 24 h following social defeat or control condition procedure. Grey bars–control rats; Red bars–defeated rats. All data presented as mean ± SEM. Unpaired two-tailed *t*-test or Mann–Whitney U test was used, * *p* = 0.0427.

**Figure 4 biomolecules-13-00035-f004:**
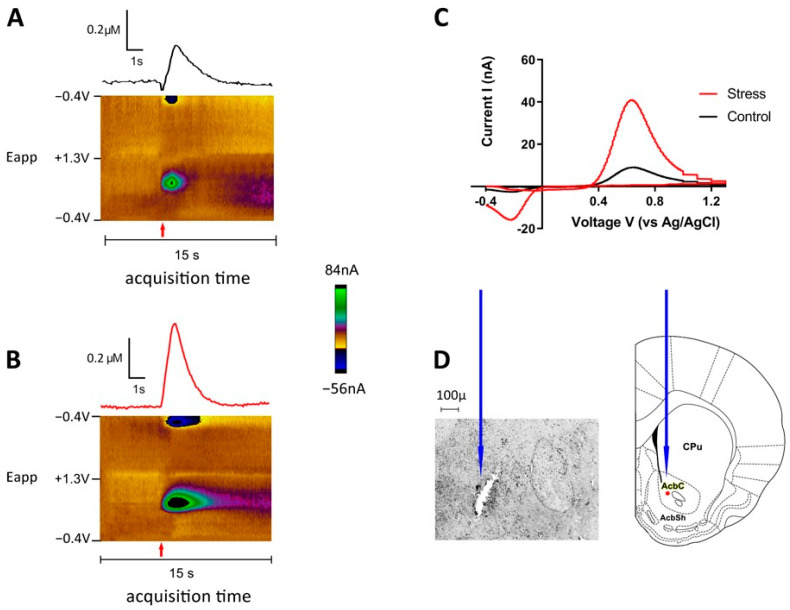
Dopamine release as measured by FSCV in the nucleus accumbens during electrical VTA stimulation. Representative color plots and traces of electrically evoked dopamine release in the brain of defeated (**A**) and control (**B**) animals are shown with time of acquisition along the *X*-axis, applied potential (Eapp) along the *Y*-axis, and background-subtracted faradaic current in pseudo-color shown in the *z*-axis. Red arrows indicate the onset of electrical stimulation (1 s, 60 Hz, 60 pulses, current 330 µA). (**C**) Representative cyclic voltammogram showing oxidation and reduction peaks at ~ 0.6 and ~ −0.2 V (respectively), confirming the signal as dopaminergic. (**D**) Histological verification of working electrode placement in the NAc as shown by a blue arrow. (Right) Electrode path observed on a coronal section 1.60 mm from Bregma [51]; (Left) tissue damage indicating placement of electrode was observed on a coronal tissue section. CPu–Caudate Putamen; Acbc-nucleus accumbens, core; AcbSh-nucleus accumbens, shell.

**Figure 5 biomolecules-13-00035-f005:**
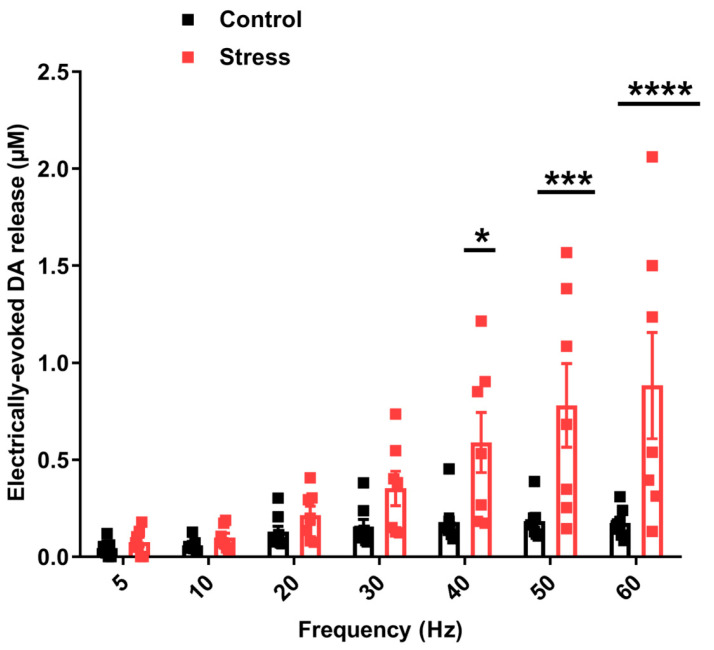
Electrically evoked dopamine is released in a frequency-dependent manner 24 h after social defeat stress exposure. Dopamine efflux in the NAc of defeated (n = 7) and control (n = 8) rats was recorded following VTA electrical stimulation (1 s, 5–60 Hz, 5–60 pulses, current 330 µA). Stimulations were taken every 10 min. Data are presented as mean ± SEM, repeated measures two-way ANOVA; * *p* < 0.05; *** *p* 0.001; **** *p* < 0.0001.

**Figure 6 biomolecules-13-00035-f006:**
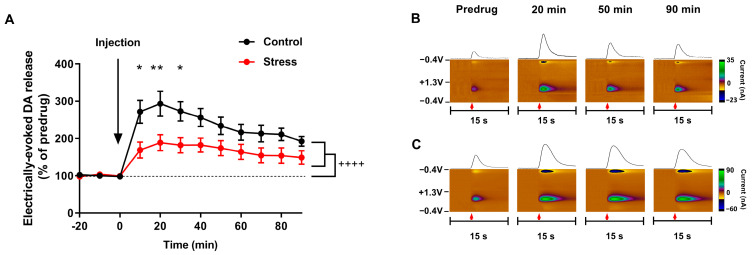
Raclopride (2 mg/kg) had a significantly reduced effect on electrically evoked accumbal dopamine release in rats exposed to social defeat stress. (**A**) Dopamine efflux following electrical stimulations in the VTA after raclopride administration. Data are presented as mean ± SEM. Control group (n = 8), defeated group (n = 5). Repeated measures two-way ANOVA, * *p* < 0.05; ** *p* < 0.01; ++++ *p* < 0.0001. (**B**,**C**) Representative color plots and traces of electrically evoked dopamine release in the brain of anesthetized control (**B**) and socially defeated (**C**) rats. Red arrow indicates the onset of electrical stimulation (1 s, 60 Hz, 60 pulses, current 330 µA).

**Figure 7 biomolecules-13-00035-f007:**
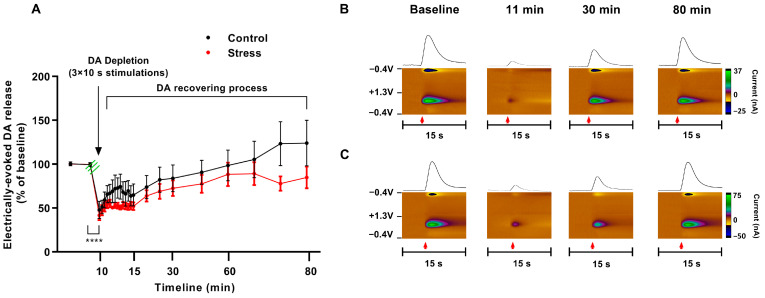
Social defeat stress does not affect dopamine depletion or subsequent release recovery. (**A**) Electrically evoked accumbal dopamine response taken 1 min (14 stimulations), 5 min (3), and 10 min (5) intervals following application of a series of stimulation pulses designed to deplete dopamine in the stimulated cells (3 × 10 s, 330 µA, 600 Hz, 600 pulses; onset of depletion stimulation pattern indicated by green lines) as a percent of pre-depletion baseline. Data presented as mean ± SEM; (n = 6 per group); **** *p* < 0.0001 (Repeated measures two-way ANOVA). (**B**,**C**) Representative color plots and traces of electrically evoked dopamine release in the brain of control (**B**) and defeated (**C**) rats taken 1, 20 and 70 min after dopamine depletion. Red arrow–onset of electrical stimulation.

## Data Availability

Not applicable.

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
