# Peer review of "Short-Term Consequences of Single Social Defeat on Accumbal Dopamine and Behaviors in Rats"

_biomolecules, 2022, doi:10.3390/biom13010035_

Round 1

Reviewer 1 Report

This is definitely an interesting study with experiments carefully designed and reflecting a good amount of work. The paper is very well written and the results are clear. Overall, the manuscript is a novel and solid paper deserving of publication in this journal. The data presented in the manuscript may provide a novel perspective for better understanding of the modulation of dopamine dynamics in stress-related responses.

This reviewer has no negative comments on this manuscript and recommends its acceptance. However, before granting the publication the authors should fix something:

1)    Pag. 1, line 38…the sentence: ’Insights into the neurochemical responses to stress can be gain from valid animal models’ should read ‘Insights into the neurochemical responses to stress can be gained from valid animal models’.

2)    Line 323, the sentence: ‘In the current study this consequence of social defeat was parallel with changes in 323 the accumbal dopamine release…’ should read ‘In the current study this consequence of social defeat was paralleled with changes in 323 the accumbal dopamine release…’

3)    Line 343, the sentence: ‘Perhaps, this consequence may disrupt normal dopamine dynamics in the nucleus accumbens and therefore endorsing some maladaptive behaviors’. The authors should be less vague when referring to ‘maladaptive behaviors’ and illustrate what they mean.

4)    Line 350, the sentence: ‘In fact, the increases in AMPA/NMDA ratio were presented acute stress and have been observed for at least 24 hours’ is not clear to me..what does it mean …were presented acute stress. Please clarify.

5)    The dose of raclopride 2 mg/kg is quite high: the authors should explain the rational of employing such a high dose.

6)    In the title , I would suggest to change ‘…social defeat for accumbal dopamine dynamics in rat….’ with ‘…social defeat on accumbal dopamine dynamics in rat….’

Author Response

Reviewer 1

  1. The spelling mistake was corrected (page 1).
  2. The error was fixed (page 11).
  3. As the reviewer requested, “maladaptive behaviors” was replaced with “depressive-like behaviors” (page 11).
  4. As the reviewer suggested, we clarified the sentence (page 11). In the current version it is “In fact, the increase in AMPA/NMDA ratio was triggered within 2 hours of acute stress and was continued for at least 24 hours”.
  5. We agree with the reviewer’s comment that the dose of raclopride (2 mg/kg) can be relatively high. Indeed, it is a really high dose in regards to behavioral parameters. However, we used this dose for neurochemical studies on anesthetized animals. We provided the rational for the use of this dose (page 3). “This raclopride dose robustly affects dopamine transmission in a way that allows for reliable detection of increased electrically-evoked dopamine release with FSCV (Park et al., 2015; Deal et al, 2021).”
  6. Based on the reviewer comment, the title was corrected.

Reviewer 2 Report

This is yet another nice piece of work that has resulted in a significant paper from a group focusing on innovative voltammetric measurements of brain monoamines that has provided so much insight into the role of dopamine in social defeat and aggression. This particular study used voltammetric, pharmacological and behavioral methods to explore the role of dopamine neuron firing in an immediate physiological response following 24-h period after social defeat. In addition potential role of these processes in the development of PTSD- and depressive-like syndrome were evaluated. The experiments are well designed and accurately performed, a broad range of techniques was used to address the question of interest. The data presented are reliable, and important for the field. They are properly evaluated and discussed. The conclusions are based on the findings. Therefore, in my opinion the manuscript is a novel and solid paper deserving of publication in this journal. I do not have any issues regarding this work. However, I would suggest some small, minor changes for improvement.

1.In my opinion, the first sentences of the Introduction section need to be referenced, even though they sum some kind of  commonly accepted knowledge.

2. This paper would benefit from a Figure with Study flow that would sum up a general scheme of this complex work.

3. The authors have performed quite extended behavioral analysis of socially defeated rats showing the increase of floating behavior in the forced swim test that is commonly interpreted as a manifestation of helplessness and depressive-like behavior in rodents. However, this important finding is not mirrored by the ms Title. I think that this an important observation that deserves more elaborated discussion in the paper. For example, a link between altered dopamine turnover and floating behavior was recently reported in mice (https://doi.org/10.1016/j.pnpbp.2020.110155)

4. In connection to a previous question, are PTSD/- depressive like changes in socially defeated rats are believed to be related to altered dopaminergic responses? If yes, can the administration of raclopride used in this study, interfere with these behaviors?

5. The presentation of sucrose test results rises a question of whether or not the calculation was based on a percent of consumed sucrose solution from total amount of consumed liquid normalized to 100% as it is indicated in materials and Methods section? Currently presented values exceed that value.

6. As for the methodology of this test, 10% sucrose concentration is a way too high concentration at which it is problematic to stratify animals by their behavioral response. Yet the variability in this test was seemingly quite high suggesting potential confounds in the protocol.

7. The paper would benefit from more rigorous description of the tests (lighting conditions and sizes of apparatuses need to indicated), as well as from better defined terms of behaviors that was expressed by the rats.

8. Did the authors consider a correlation analysis between floating and other behaviors, e.g., rearing scores, and voltammetric read-outs?

9. It is not very clear why group sizes were different across the number of assays used.

Author Response

Reviewer 2

  1. As the reviewer suggested, the first sentences of the introduction were referenced (page 1).
  2. Based on the reviewer’s comment, we added the figure with a general scheme of our complex study (see Figure 1A).
  3. We agree with the reviewer’s criticism that the title of the manuscript did not reflect a behavioral aspect of our study. Therefore, we modified the title accordingly (“Short-term consequences of single social defeat on accumbal dopamine and behaviors in rats”). The latest finding, which revealed a link between altered dopamine turnover and floating behavior, was acknowledged in the discussion (page 11).
  4. Our study demonstrated that depressive like changes in socially defeated rats are associated with altered dopamine response in the nucleus accumbens. Whether a causal relationship exists between these behavioral and neurochemical changes should be proved in future studies. Certainly, the administration of raclopride should interfere with these behaviors. However, it is not the case for our study, since behaviors and effects of raclopride were explored on separate groups of rats. We apologize for being unclear in the previous version of our manuscript. The new figure (1A) clarifies this fact.
  5. The reviewer is right about a misrepresentation of sucrose test results. It was corrected (Figure 3A).
  6. High sucrose concentrations were used in many previous studies on stress (for example see Wright RL, Gilmour G, and Dwyer DM, 2020). However, we agree with the reviewer’s concern that this concentration can be relatively high to stratify animals by their behavioral response. We will further explore the consequence of single social defeat on sucrose intake using a range of the concentrations in future studies.
  7. Additional details on behavioral tests were provided (2.1. Animal and behavioral tests, page 2).
  8. We did not consider a correlation analysis between behavioral changes and voltammetric recordings, since these experiments were performed on different groups of rats. Therefore, it is technically inappropriate.
  9. The group size for the certain assay was based on previous experience with these tests. This reflects our effort to minimize suffering and reduce the number of rats in this study, while applying an appropriate statistical analysis. We indicated this fact in the Materials and Methods (2.1. Animal and behavioral tests).

Reviewer 3 Report

The authors of the present study explore the consequences of a single social defeat on Nucleus accumbens dopamine release evoked by an electrical stimulation of the VTA measured by a fast-scan cycli voltammetry (FSCV) using rats as an animal model.

I have some comments that hope will be helpful to improve de manuscript.

Introduction: Lines 57-62. These data are provided by other studies. The way it is written seems it has been performed by the authors or is a conclusion that the authors have made. Please re-write these sentences.

Materials and methods section:

In this section, the authors described how they proceeded with the social defeat experiment. However how the control condition was performed is not totally clear.

Please add the control condition to figure 1 where the intruder rat is shown but the control rat is missing.

In lines 91-92 the authors state that during the 10-min period in where the protective inset cage is removed from this test a Sprague-Dawley rat was introduced to its counterpart for 20-min in order to avoid any aggressive confrontations. Is this the control rat condition? Is it also a male rat? Why is there not competition between these rats? Is it because they are from the same strain? Please, add more information.

Open field experiment. Could the authors include in the presented figure 3 the time (s) or the % of time that the control vs. the stressed rats remained in the center vs. the periphery of the open field. This would add information in regards to the anxiety behavior of these animals after the SD protocol.

In figure 3, the authors show that the control group performs more grooming behavior that the stressed group. However, this aspect is not discussed later in the discussion section. Please add a possible explanation regarding this change.

Also in this figure, I would suggest the authors to move figure 3B to the end as 3F as it is ordered in the text. This way is easier for the reader to follow the text and the figures.

In Figure 6 B and C. Could the same scale line (Current, nA) be used as in figure 4? The same for figure 7 B and C.

Discussion:

Lines 365-366. “Consequently, the reduced inhibition of dopamine release via activation of these receptors could lead to the enhanced dopamine due to diminished level of endogenous agonist and contrary wise”.This statement is not clear. Where is dopamine enhanced after the activation of kappa opioid receptors? In the Nac?

Line 394. I believe in the context of the manuscript “naïve” should be changed to “control” rats.  

In the ending of the discussion (lines 408-410) section the authors hypothesize that since raclopride and dopamine compete for the D2 receptors,  increased electrically-evoked dopamine could modify the pharmacological effect of the drug, weakening presynaptic autoreceptor blockade and therefore down-regulating the release. Are they referring to presynaptic autoreceptor blockade in VTA neurons?

Other considerations: use italics for -ex vivo preparation lines 356, 368- ad libitum line 81.

Author Response

Reviewer 3.

  1. As the reviewer requested, the sentence was rewritten (page 2).
  2. Following the reviewer’s comment, we added the control condition (social interaction) to figure 1 (see Figure 1A).
  3. The reviewer is correct. It is the control rat condition (“In a separate set of experiments, a SD rat was introduced to its counterpart….”). We clarified this fact (page 2). The requested information was provided (page 2).
  4. Based on the reviewer’s suggestion, we added the time that animals remained in the center of the open field (Figure 3 C).
  5. There is no difference in the grooming between control and stressed groups in figure 3 E. This figure reports results on behavioral consequences of single social defeat and this is together with the data on dopamine release is the focus of this study. Behavioral changes, which were observed during the social defeat procedure, were characterized and discussed in many previous studies. We reported these changes in order to confirm a social defeat condition.
  6. We moved figure 3B to the end as 3F, as it was proposed by the reviewer (see Figure 3).
  7. The scales should be individual for different color plots.
  8. The reviewer’s assumption is correct. Dopamine release is changed in the nucleus accumbens. We clarified it (page 12).
  9. As the reviewer requested, “naïve” was changed to “control” (page 12).
  10. The reviewer is right. We were referring to presynaptic autoreceptor blockade in the VTA. We clarified this statement (page 13).
  11. We changed “ex vivo” and “ad libitum” to italics.

Round 2

Reviewer 3 Report

The authors have greatly improved the manuscript.

However I have noticed that in the revised version of figure 3 the graph of 3A sucrose/water ratio% is different from the original version. The result does not change since there are no statistical differences between the ¨control¨ and ¨defeated¨ rats. Why has the data of this figure been modified?

Author Response

The data presentation of the figure 3 A was modified since other reviewer raised the concern (reviewer 2, question 5) about a possible inaccuracy in the normalizing of presented values (“The presentation of sucrose test results raises a question of whether or not the calculation was based on a percent of consumed sucrose solution from total amount of consumed liquid normalized to 100% as it is indicated in materials and Methods section. Currently presented values exceed that value”). Indeed, we found a mistake in the equation that was created in Excel to obtain these results.  Therefore, the new figure reflects corrected values. Importantly, the same numbers were used in both calculations (old manuscript version and new version), whereas the incorrect equation was originally used. We apologize for this mistake in the previous version.